# Projecting heat-related excess mortality under climate change scenarios in China

Jun Yang [1,2,3,15✉], Maigeng Zhou[4,15], Zhoupeng Ren [5,15], Mengmeng Li[6], Boguang Wang[1,2,3], De Li Liu [7,8], Chun-Quan Ou[9], Peng Yin[4], Jimin Sun[10], Shilu Tong[11,12,13], Hao Wang[1,2,3], Chunlin Zhang[1,2,3], Jinfeng Wang [5], Yuming Guo [14] & Qiyong Liu[10✉]

Recent studies have reported a variety of health consequences of climate change. However, the vulnerability of individuals and cities to climate change remains to be evaluated. We project the excess cause-, age-, region-, and education-specific mortality attributable to future high temperatures in 161 Chinese districts/counties using 28 global climate models (GCMs) under two representative concentration pathways (RCPs). To assess the influence of population ageing on the projection of future heat-related mortality, we further project the age-specific effect estimates under five shared socioeconomic pathways (SSPs). Heat-related excess mortality is projected to increase from 1.9% (95% eCI: 0.2–3.3%) in the 2010s to 2.4% (0.4–4.1%) in the 2030 s and 5.5% (0.5–9.9%) in the 2090 s under RCP8.5, with corresponding relative changes of 0.5% (0.0–1.2%) and 3.6% (−0.5–7.5%). The projected slopes are steeper in southern, eastern, central and northern China. People with cardiorespiratory diseases, females, the elderly and those with low educational attainment could be more affected. Population ageing amplifies future heat-related excess deaths 2.3- to 5.8-fold under different SSPs, particularly for the northeast region. Our findings can help guide public health responses to ameliorate the risk of climate change.

[1] Institute for Environmental and Climate Research, Jinan University, Guangzhou, China. [2] Guangdong-Hongkong-Macau Joint Laboratory of Collaborative Innovation for Environmental Quality, Guangzhou, China. [3] JNU-QUT Joint Laboratory for Air Quality Science and Management, Jinan University, Guangzhou, China. [4] National Center for Chronic and Noncommunicable Disease Control and Prevention, Beijing, China. [5] State Key Laboratory of Resources and Environmental Information System (LREIS), Institute of Geographic Sciences and Nature Resources Research, Chinese Academy of Sciences, Beijing, China. [6] State Key Laboratory of Oncology in Southern China, Department of Epidemiology, Cancer Prevention Center, Sun Yat-Sen University Cancer Center, Guangzhou, China. [7] NSW Department of Primary Industries, Wagga Wagga Agricultural Institute, Wagga Wagga, NSW, Australia. [8] Climate Change Research Centre, University of New South Wales, Sydney, NSW, Australia. [9] State Key Laboratory of Organ Failure Research, Department of Biostatistics, Guangdong Provincial Key Laboratory of Tropical Disease Research, School of Public Health, Southern Medical University, Guangzhou, China. [10] State Key Laboratory of Infectious Disease Prevention and Control, Collaborative Innovation Center for Diagnosis and Treatment of Infectious Diseases, National Institute for Communicable Disease Control and Prevention, Chinese Center for Disease Control and Prevention, Beijing, China. [11] Shanghai Children's Medical Center, Shanghai Jiao Tong University, Shanghai, China. [12] School of Public Health and Institute of Environment and Population Health, Anhui Medical University, Hefei, China. [13] School of Public Health and Institute of Health and Biomedical Innovation, Queensland University of Technology, Brisbane, Australia. [14] Department of Epidemiology and Preventive Medicine, School of Public Health and Preventive Medicine, Monash University, Melbourne, Australia. [15] These authors contributed equally: Jun Yang, Maigeng Zhou, Zhoupeng Ren. ✉email: yangjun_eci@jnu.edu.cn; liuqiyong@icdc.cn

Global climate change has become one of the most challenging environmental issues[1]. The most immediate and direct consequence of climate variability and change on public health is the stable increase in global surface temperature, accompanied by the enhanced frequency, severity, and duration of heat waves[2]. For instance, in the summer of 2003 European heatwave caused more than 70,000 excess deaths[3]. An extreme heatwave in Moscow and Western Russia during June–August 2010 led to over 55,000 additional deaths[4]. Considering the ever-worsening situation, there is an urgent need to protect the health burden of future temperature increases to support the development of adaptation strategies and resource allocations.

Previous studies have documented that people with chronic diseases, especially cardiopulmonary diseases, are more susceptible to high temperature[5–7]. To date, current investigations have focused on either the effect of historical temperature on different diseases or projections of specific diseases due to future temperatures over a short-time period[7–9], neither of which can clearly reveal the long-term trend of temperature-related health burdens of different diseases. In addition, current studies have been mainly confined to projecting the mortality effects of climate change in highly developed cities, leading to considerable uncertainties when generalizing the evidence to larger regions with diverse socioeconomic levels[8,10,11]. These previous investigations have different analytical strategies, model specifications, and study periods, which may reduce the comparability of results across regions; therefore, they cannot provide an assessment of vulnerability at regional levels or a whole picture of the development of climate change mitigation initiatives for policymakers.

Older adults are particularly sensitive to climate change[6]. The vulnerability of the elderly is associated with physiological and social factors, such as living alone, pre-existing chronic diseases, reduced physiological function in thermoregulation to heat stress and limited access to medical care and cooling household appliances. Driven by lower fertility rates but longer life expectancies, population aging is accelerating globally at an unprecedented rate[12]. Thus, the increase in vulnerable populations caused by population aging may amplify future heat-related health burdens. However, changes in the population structure have not been carefully considered in previous studies[13–15], and thus they may underestimate the harmful impact of climate change on human health.

China, the most populous developing country and among the counties with the most rapidly aging population, represents a typical example for assessing the increases in the susceptibility of the elderly to future climate and demographic changes[16]. However, quantitative evidence on the impact of future high temperatures on health in China is scarce[7,11,13], and the influence of future population aging on heat vulnerability often is ignored in these studies.

Here, we assess the excess mortality risk of future heat exposure by cause of death, region, and individual characteristics in 161 Chinese districts/counties (Supplementary Fig. 1 and Supplementary Table 1) under different future weather scenarios using 28 general climate models (GCMs) (Supplementary Table 2). We find that people living in southern, eastern, central, and northern China, subjects with cardiorespiratory diseases, females, the elderly, and the illiterate will suffer more adverse effects from future high temperatures in China. Population aging will remarkably amplify heat-related deaths in the future.

## Results

**Descriptive data**. Figure 1 presents the annual mortality rate (per million population) and average daily mean temperature (°C) during 2007–2013 in 161 Chinese districts/counties. A total of

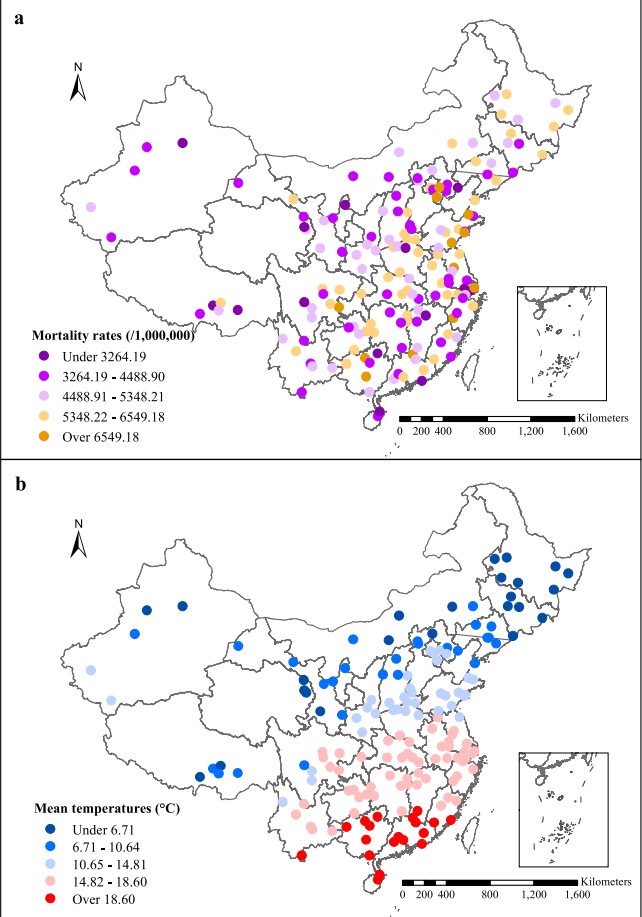

**Fig. 1 The annual mortality rate (per million populations) and annual mean temperature (°C) in 161 Chinese disease surveillance points.** The colors represent different ranges of **a** annual mortality rates and **b** average daily mean temperatures during 2007–2013. Source data are provided as Supplementary Data 1.

2,742,717 non-accidental deaths, 1,274,558 cardiovascular deaths, 670,017 stroke deaths, 431,814 ischemic heart disease deaths, 397,738 respiratory deaths, and 302,125 chronic obstructive pulmonary diseases (COPD) deaths were obtained from 7 regions in mainland China during the study period. The non-accidental mortality rate ranged from 4055 per million population in the Northwest to 5322 per million population in the northeast, and the average annual mean temperature increased from 6.7 °C in the northeast to 22.7 °C in the south. Descriptive information on the number of deaths by cause and individual characteristics is presented in Supplementary Table 3.

**Temporal and spatial trends in temperature**. Figure 2 shows the trends in historical (1960–2005) and future projected temperatures (2006–2099) by region under the RCP4.5 and RCP8.5 scenarios. A steep increase in projected temperature is consistently observed across this century under the RCP8.5 scenario, while only a slight increase in projected temperature is assumed for RCP4.5 after the middle of this century. By the end of the 21st century, the average temperature will rise 1.5 °C under RCP4.5 and 3.8 °C under RCP8.5. The southern, eastern, and central regions will have higher absolute temperatures in the 2090s, while more rapidly increasing magnitudes of mean temperature and higher temperature variability are observed in the northeast, northern and northwest regions (Supplementary Table 4).

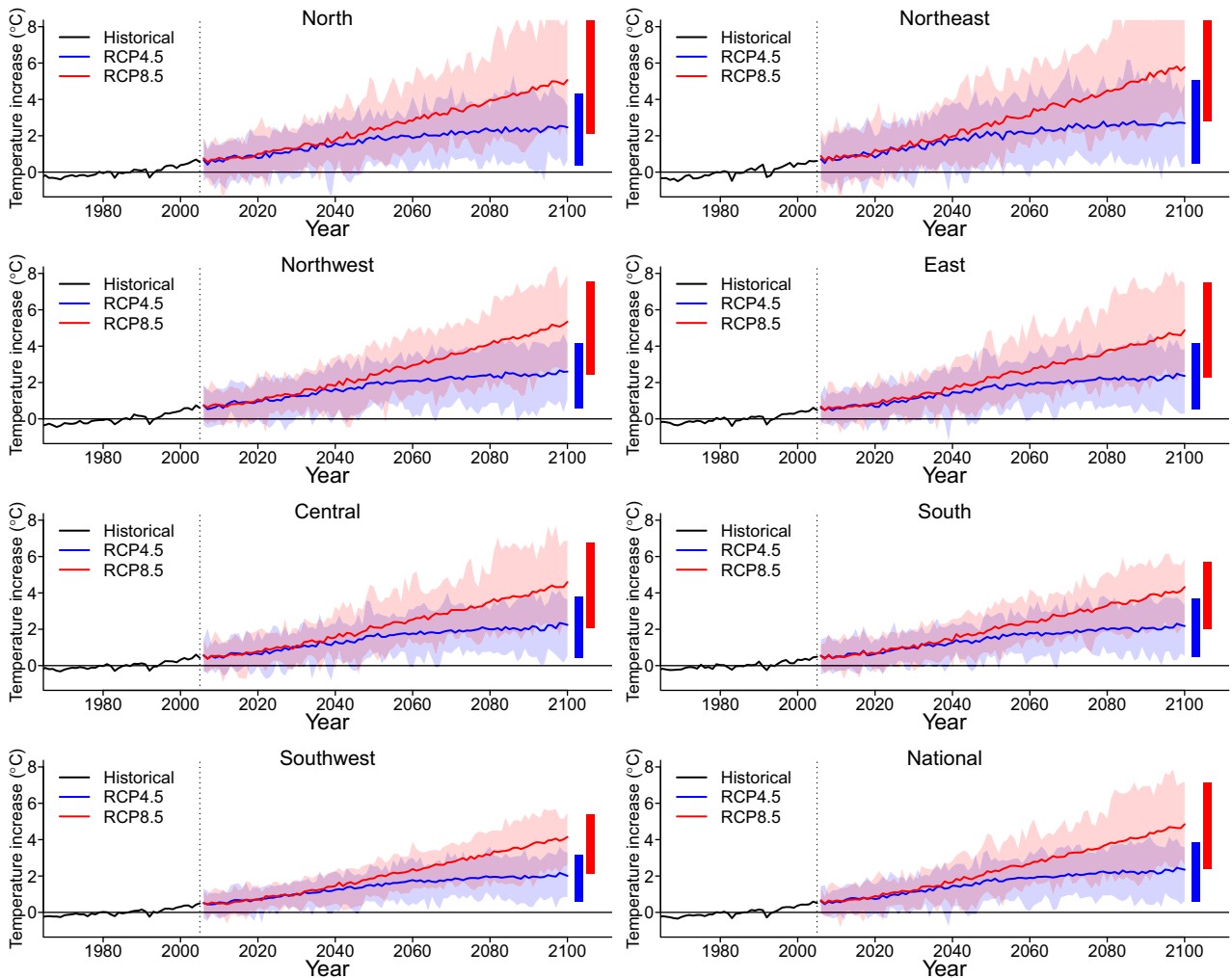

**Fig. 2 Temporal trends in projected temperature in China.** RCP4.5 and RCP8.5 represent the low and high emission scenarios, respectively. Solid lines denote the mean annual temperature estimated across the 28 general circulation models. The shaded areas denote the temperature range for each year. The two vertical bars on the right show the average annual minimum and maximum for each projected temperature series. Source data are provided as Supplementary Data 2.

**Baseline temperature–mortality association**. Supplementary Fig. 2 presents the relationships between ambient temperature and non-accidental mortality across lag 0–14 days by region. Generally, the temperature–mortality associations were U- or reverse J-shaped. Cochran's $Q$ test and $I^2$ statistic indicate significant between-city heterogeneity across 161 districts/counties ($Q = 1141.9$, $I^2 = 44.7\%$, $P < 0.001$). We observed higher heat-related health risks for northern, central, and southwestern China. The overall cumulative exposure–response curves for different mortality categories, genders, age groups, and educational attainments are presented in Supplementary Fig. 3.

**Future heat-related excess mortality by region and individual characteristics**. Table 1 presents the estimated attributable fraction of deaths due to future high temperatures under the RCP4.5 and RCP8.5 scenarios by region with the demographic structure remaining constant at the level of 2010s. We projected a steep increase in heat-related excess mortality from 1.9% (95% eCI: 0.2–3.3) in the 2010s to 2.4% (95% eCI: 0.4–4.1) in the 2030 s, 3.2% (95% eCI: 0.6–5.6) in the 2050s, and 5.5% (95% eCI: 0.5–9.9) in the 2090s under RCP8.5. Compared to the 2010s, in the 2030s, 2050s, and 2090s, the relative changes are 0.5% (95% eCI: 0.0–1.2), 1.3% (95% eCI: −0.1 to 3.0) and 3.6% (95% eCI: −0.5 to 7.5), respectively.

Whereas the gradient gradually increased under RCP4.5, rising from 1.8% (95% eCI: 0.2–3.3) in the 2010s to 2.2% (95% eCI: 0.4–3.9) in the 2030s, 2.6% (95% eCI: 0.5–4.6) in the 2050s and 3.1% (95% eCI: 0.6–5.5) in the 2090s, with corresponding relative changes of 0.4% (95% eCI: 0.0–1.0), 0.8% (95% eCI: 0.0–2.0) and 1.2% (95% eCI: 0.0–3.1), respectively. The attributable numbers of heat-related deaths are 76,364 (95% eCI: 7670–136,841) in the 2010s, and 128,346 (95% eCI: 24,051–232,185) and 228,728 (95% eCI: 22,593–414,619) in the 2090s under the RCP4.5 and RCP8.5 scenarios, respectively (Supplementary Table 5). In addition, important differences were observed among regions. The projected slopes are much steeper in the southern, eastern, central, and northern regions under RCP8.5, with estimates of 6.2–6.5% in the 2090s relative to 1.8–2.9% in the 2010s. The differences in excess mortality by region are provided in Supplementary Table 6. Furthermore, when projecting the estimate by the level of gross domestic product (GDP) per capita, relatively higher heat-related excess mortality is observed in districts or counties with low GDP per capita during the baseline period, while the projected slopes are steeper in districts or counties with high GDP per capita (Supplementary Table 7).

For different subgroups, subjects with cardiovascular and respiratory diseases, females, those aged over 75 years old and the

**Table 1 Heat-related excess mortality (%, 95% eCI) by region, period, and climate change scenario, assuming no adaptation or population changes.**

| Region | RCP 4.5 | | | | RCP 8.5 | | | |
|---|---|---|---|---|---|---|---|---|
| | 2010s | 2030s | 2050s | 2090s | 2010s | 2030s | 2050s | 2090s |
| Northern | 2.8 (−0.9 to 6.1) | 3.2 (−0.7 to 6.6) | 3.6 (−0.4 to 7.2) | 4.0 (−0.2 to 7.8) | 2.9 (−0.9 to 6.2) | 3.3 (−0.6 to 6.8) | 4.2 (−0.0 to 7.9) | 6.4 (1.1–12.0) |
| Northeast | 0.9 (0.1–1.6) | 1.2 (0.1–2.3) | 1.6 (0.1–3.2) | 1.9 (0.1–4.1) | 0.9 (0.1–1.8) | 1.3 (0.1–2.5) | 2.0 (0.2–4.2) | 4.3 (0.4–8.9) |
| Northwest | 0.5 (−0.6 to 1.5) | 0.7 (−0.8 to 2.0) | 0.9 (−1.1 to 2.6) | 1.1 (−1.3 to 3.3) | 0.5 (−0.6 to 1.5) | 0.8 (−0.9 to 2.2) | 1.1 (−1.5 to 3.5) | 2.3 (−3.5 to 7.0) |
| Eastern | 1.8 (0.6–2.9) | 2.3 (0.9–3.6) | 2.8 (1.1–4.6) | 3.3 (1.3–5.7) | 1.8 (0.6–3.0) | 2.4 (1.0–3.9) | 3.5 (1.5–5.6) | 6.2 (2.5–10.2) |
| Central | 2.6 (−0.4 to 5.5) | 3.1 (−0.1 to 6.0) | 3.5 (0.2–6.6) | 4.0 (0.5–7.2) | 2.7 (−0.4 to 5.5) | 3.2 (0.0–6.2) | 4.0 (0.6–7.3) | 6.3 (1.8–10.6) |
| Southern | 1.9 (−0.5 to 3.7) | 2.3 (−0.7 to 4.6) | 2.9 (−1.2 to 6.2) | 3.4 (−1.6 to 7.7) | 1.9 (−0.5 to 3.7) | 2.5 (−0.8 to 5.1) | 3.6 (−1.8 to 7.8) | 6.5 (−5.8 to 14.8) |
| Southwest | 1.2 (−0.4 to 2.5) | 1.4 (−0.7 to 3.2) | 1.7 (−1.1 to 4.1) | 2.1 (−1.4 to 5.1) | 1.1 (−0.4 to 2.5) | 1.5 (−0.8 to 3.4) | 2.1 (−1.6 to 5.2) | 3.9 (−4.2 to 10.1) |
| National | 1.8 (0.2–3.3) | 2.2 (0.4–3.9) | 2.6 (0.5–4.6) | 3.1 (0.6–5.5) | 1.9 (0.2–3.3) | 2.4 (0.4–4.1) | 3.2 (0.6–5.6) | 5.5 (0.5–9.9) |

A distributed lag nonlinear model was used to estimate the district/county-specific temperature–mortality association with 14 days of lag adjusted for time trends and day of the week, which were pooled in a multivariate meta-analysis. Then, estimates of the attributable fraction of deaths due to high temperature, defined as temperatures above the optimum temperature, were calculated by the regional and national levels. Monte Carlo simulations generating 1000 samples were computed to produce empirical confidence intervals.

illiterate will suffer from higher adverse impacts of future high temperatures (Fig. 3). For instance, the estimated heat-related attributable fractions in the 2090s under RCP8.5 are 6.3% (95% eCI: 2.3–10.2) for cardiovascular mortality and 7.7% (95% eCI: −1.6 to 14.7) for respiratory mortality; 4.8% (95% eCI: 1.3–8.5) and 6.5% (95% eCI: 1.3–10.9) for males and females, respectively; 4.6% (95% eCI: 0.8–8.0) and 6.6% (95% eCI: 2.5–10.8) for the youth and the elderly, respectively; 6.4% (95% eCI: −0.5 to 12.1) and 5.3% (95% eCI: 2.0–8.7) among the illiterate and persons with higher educational attainment, respectively (Supplementary Table 8). The attributable numbers of deaths by individual characteristics are provided in Supplementary Table 9, and the differences in excess mortality compared with that in the 2010s are shown in Supplementary Table 10.

**Population aging on heat-related excess mortality.** Figure 4 presents the changes in the number of deaths attributable to high temperature for different age groups under the six population growth scenarios in the 2030s, 2050s, and 2090s. Remarkably, population aging will amplify future heat-related additional deaths of those aged 75 years and above. For instance, the heat-related excess mortality will increase to 438,899–913,986 for the elderly in RCP8.5 in the 2090s under the five shared socio-economic pathway (SSPs), with the highest estimate being 913,986 (95% eCI: 346,237–1,493,616) under SSP5, compared to 133,711 (95% eCI: 50,653–218,508) under no population change scenario. For different regions, the fastest increasing rate driven by aging was observed in the northeast region, with 67,875–127,489 heat-related excess deaths for the elderly under five SSPs relative to 11,464 (95% eCI: 1749–21,893) under no population change scenario (Supplementary Figs. 4 and 5; Supplementary Tables 11 and 12). However, the number of deaths attributable to future high temperatures for the youth consistently decreased across the five SSPs at the regional and national levels, particularly for the SSP1, SSP4, and SSP5 scenarios.

**Sensitivity analyses.** Multiple analyses were conducted to test the robustness of the main results. Effect estimates were very similar when we used 6-10 df for the time trends and different maximum lag days for the temperature–mortality association, respectively. The estimates remained stable after separately adjusting for other meteorological factors, including relative humidity and wind speed, and for simultaneous exposure to air pollutants (Supplementary Table 13).

## Discussion
Our study finds that the projection of heat-related excess mortality increases by decade and RCP in China, particularly under the RCP8.5 scenario. Future high temperatures in the 2030s, 2050s, and 2090s will cause larger numbers of heat-related deaths among people living in the southern, eastern, central, and northern areas, and among those with cardiorespiratory diseases, females, the elderly, and the illiterate.

Our estimates show that heat-related mortality will escalate in all regions under different climate change scenarios. Nationally, the attributable fraction of non-accidental mortality due to high temperatures will increase from 1.9% in the 2010s to 5.5% in the 2090s under RCP8.5, which is comparable to the estimates in another study based on 15 Chinese cities (1.0% in the 2010s and 6.1% in the 2090s)[13]. A larger increase in heat-related deaths was noted for the pathway of high future $CO_2$ emissions (RCP8.5) in comparison to the stabilizing forced scenario (RCP4.5). These findings are broadly concordant with those from previous studies on different regions and populations[9,15,17]. However, direct comparison of our effect

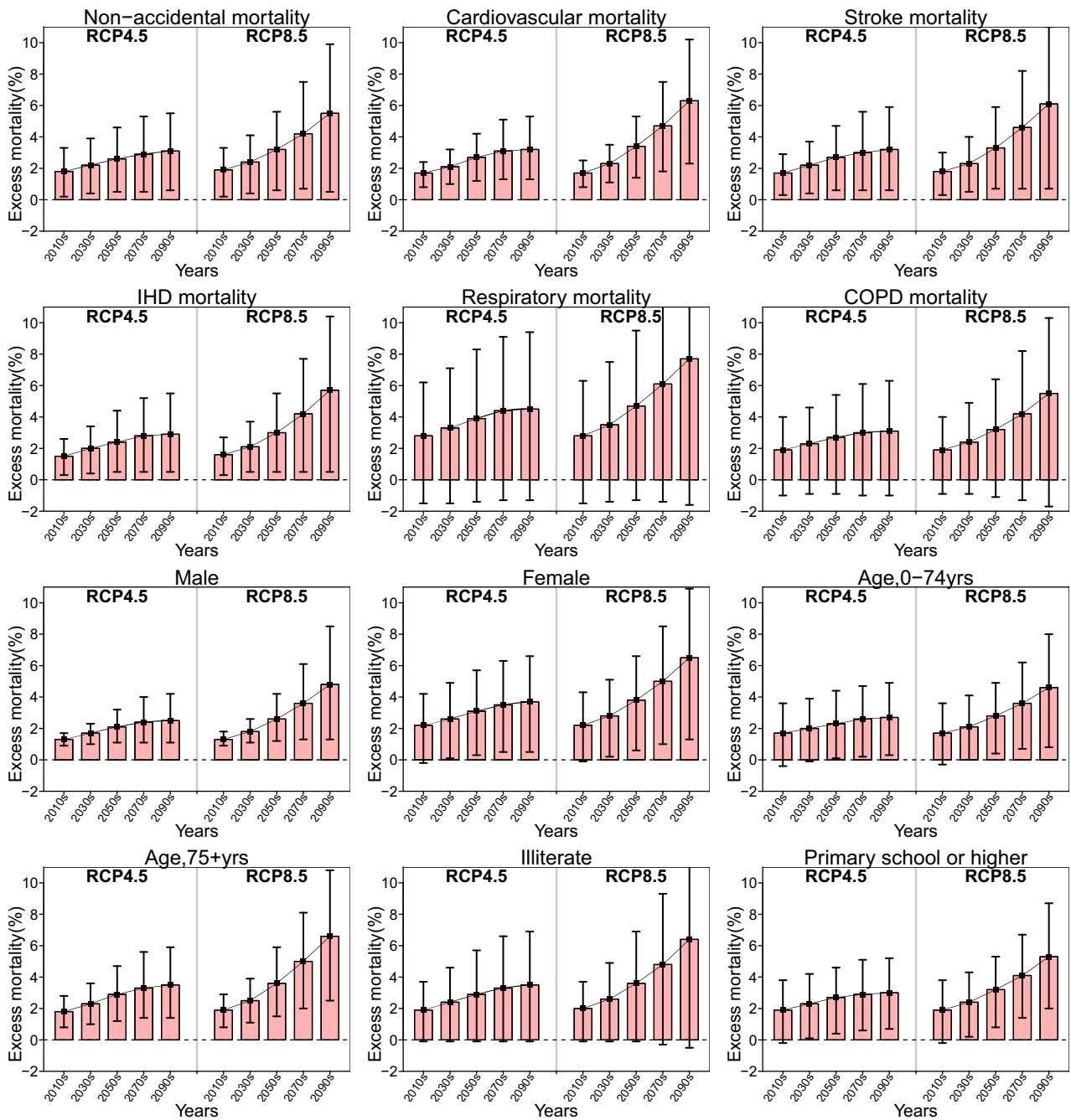

**Fig. 3 The trends in heat-related excess mortality by period, cause, and individual characteristics in China assuming no adaptation or population changes.** The excess mortality (red bar) was represented as the fraction of deaths (%) attributable to high temperature, with 95% empirical confidence intervals (the vertical black line). A distributed lag nonlinear model was used to estimate the district/county-specific temperature-mortality association with 14 days of lag adjusted for time trends and day of the week, which were pooled in a multivariate meta-analysis. Then, estimates of the attributable fraction of deaths due to high temperature, defined as temperatures above the optimum temperature, were calculated at the national level. Monte Carlo simulations generating 1000 samples were computed to produce empirical confidence intervals. The analyses were separately repeated by cause of death and individual characteristics. Source data are provided as Supplementary Data 3.

estimates to those in previous studies is difficult because of various climate models, population growth scenarios, time periods, and downscaling approaches. For instance, Petkova and colleagues reported increases of 171.1–286.5% and 255.6–410.8% in heat-related deaths in three major cities of the United States in the 2050s under RCP4.5 and RCP8.5, respectively (the heat-related mortality rates were 8.8–17.1 per 100,000 and 11.7–16.0 per 100,000, compared to 2.9–4.5 per 100,000 during 1971–2000)[18]. In tandem with the previous findings[9,16], our study underscores that climate change

mitigation and adaptation strategies are imperative to protect public health.

Our study finds significant spatial heterogeneity in future changes in high temperatures. Regions with hotter climates, such as the southern, eastern, and central regions, are characterized by relatively stronger future high temperature-related impacts, especially under the high global warming scenario (RCP8.5). Our findings are consistent with those from a global analysis[13], which may be due to the increased frequency, intensity, and duration of

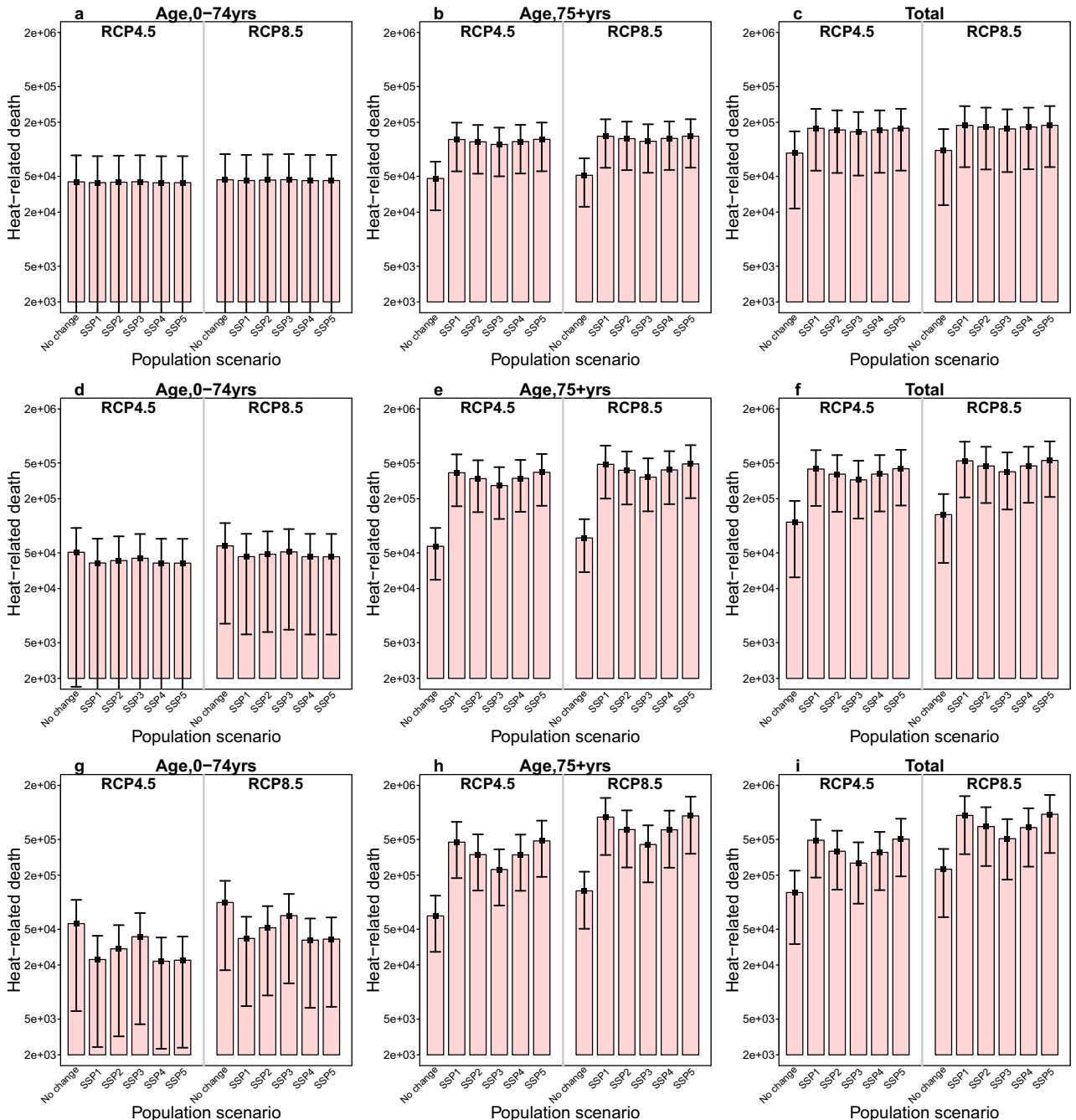

**Fig. 4 The number of heat-attributable deaths by period, age, and population scenario in China. a–c** Respectively show the number of heat-attributable deaths (red bar) in the 2030 s for different age groups under six population scenarios (no change, SSP1–SSP5). **d–f** Respectively show the number of heat-attributable deaths in the 2050s by age and population scenario. **g–i** Respectively show the number of heat-attributable deaths in the 2050s by decade, age, and population scenario. The black vertical segments represent the 95% empirical confidence interval. A distributed lag nonlinear model was used for estimating district/county-specific temperature–mortality associations with 14 days of lag adjusted for time trends and day of the week, which were pooled in a multivariate meta-analysis. Then, based on the age-specific temperature–mortality association and age-specific projected population, estimates of the attributable number of deaths due to high temperature, defined as temperatures above the optimum temperature, were calculated for different age groups under the five SSPs. Monte Carlo simulations generating 1000 samples were computed to produce the empirical confidence interval. Source data are provided as Supplementary Data 4.

extremely high temperatures in these regions, although contrasting results were also reported in the previous literature[19]. This discrepancy may be partly explained by the differences in the vulnerability of local residents to heat exposure and the GCM models used. Furthermore, we compare the effect estimates for different levels of GDP per capita under current and future

scenarios. Relatively lower heat-related mortality risk in the current decade is observed in districts/counties with higher GDP per capita, but with a steeper projected slope of heat-related excess mortality in future decades. Better medical services and a higher prevalence of air conditioning use in these areas may ameliorate the heat vulnerability of local populations.

However, rapid urbanization, in conjunction with the heat island effect, is associated with a rapid increase in temperature and may aggravate the health risk of future heat exposure in these highly developed regions (e.g., eastern China) and in districts/counties with a higher socioeconomic status[20]. Surprisingly, a high burden from future high temperatures also is found in the north, which may be due to the higher heat vulnerability level of the local residents and the relatively rapidly increasing temperature rate in this region. These findings are helpful in the design of better-localized intervention strategies to reduce the future heat-induced death burden.

Previous studies have documented that cardiovascular and respiratory diseases are particularly sensitive to ambient high temperature in comparison with all-cause or non-accidental diseases[5,21–23]. The mechanism of the heat effect on cardiovascular and respiratory pathophysiology includes changes in thermoregulation response, blood viscosity and surface blood circulation, and endothelial cell damage[24]. In the present study, the future projection of the heat-related attributable number of deaths from cardiovascular disease is remarkably greater than that from respiratory disease, with corresponding total estimates of 122,006 (95% eCI: 45,041–197,769) and 47,168 (95% eCI: −9968 to 89,509) deaths in the 2090s under RCP8.5, respectively. The larger burden of cardiovascular disease may be related to the higher baseline mortality rates. For instance, the age-standardized mortality rates of cardiovascular disease and respiratory disease in 2010 were 230.8 per 100,000 and 77.2 per 100,000, respectively. Driven by lifestyle changes, the health burdens of non-communicable diseases are predicted to increase substantially in the coming decades, particularly for cardiovascular disease and COPD[25,26]. Therefore, continuous attention needs to be directed to these diseases, and consistent epidemic surveillance is necessary.

Population aging has been viewed as an important factor that could substantially exacerbate the health burden of future rising temperatures globally. In comparison with the scenario of no population change, the five SSPs, especially SSP1 and SSP5, which assume low fertility, low-to-medium mortality, medium-to-high migration, and high education level, would result in heat-related mortality in the elderly that is 2.3–5.8 times higher by the 2090s under RCP8.5. A recent study conducted in Beijing reported a very similar estimate in the 2080s among those 65 years or older under RCP8.5, which is 264.9% greater than the estimate in the 1980s[10]. When considering both future population growth and the aging rate, the abovementioned study observed that this estimate will increase to 7.4 times[10]. Interestingly, the fastest increase in heat-related excess mortality driven by aging is detected in the northeastern region, which may be associated with the low birth rate but high population out-migration in this region[27]. Given the challenge of the rapidly aging population in China[28], our study may contribute to the development of mitigation policies and interventions to protect the elderly from exposure to current and future high temperatures, including health care planning and public health activities to increase their heat resilience. The vulnerability of the elderly to heat exposure is mainly due to a combination of impaired thermoregulatory capacity to heat, and a high prevalence of comorbidities involving the cardiovascular, respiratory, endocrine, and renal systems[24].

Females were consistently found to be disproportionately vulnerable to the effect of future temperature increases[29–31]. Potential explanations could be that some women experience lower economic independence, lower control over productive resources, higher rates of low educational levels, and lack of access to proper information[29–31]. In terms of educational level, we observed that future high temperatures will cause more heat-related deaths among people with lower educational levels. Educational attainment is closely associated with economic status.

Those with lower educational attainment are more likely to have lower-incomes, and to be unable to afford good living conditions, medical services, and household cooling appliances, which may exacerbate the risk from climate exposure.

The present study examines the mortality burden of future high temperatures using 28 climate models, two emission scenarios, and a future aging population under five SSPs. However, the uncertainty regarding urbanization, population adaptation and acclimatization, and population structure changes still need to be noted. We assumed that the shape of the exposure–response curve for the temperature–mortality association, as well as the climate adaptive capacity, would remain constant throughout the study period at the level of the 2010s. Therefore, our estimates should be interpreted as potential impacts of future high temperatures under hypothetical scenarios rather than projections of future excess mortality[13]. However, although previous studies have observed that the temperature–mortality association could change over time[9,22], the extent or rate of population adaptation is still unclear. In addition, future population acclimatization to elevated temperatures also is not considered in this study, which may result in an overestimation of the health effects of future high temperatures. However, Baccini et al.[14] argued that current evidence on the extent to which short- or long-term acclimatization influences mortality risk is limited and inconsistent. Furthermore, as an important driver of global warming, urbanization could greatly reshape the spatial distribution and population structure[20], particularly in countries with rapid urbanization, such as China, where the rate of urban population growth increased from 45% in 2007 to 60% in 2019[32]. Urban population growth, coinciding with an increasing urban heat island effect, can act as another source of uncertainty in projections of future heat-related death burdens[33]. In addition to future population aging, future demographic changes in other heat-sensitive subpopulations (e.g., females and those with low socioeconomic status) and shifts in chronic disease morbidity and mortality (e.g., cardiovascular diseases and respiratory diseases) also may partly modify the vulnerability of local people to future heat exposure. Future research is still warranted to appropriately include the aforementioned sources of uncertainty in the assessments of future heat-related health risks.

Our study has several major public health implications. First, future heat-related deaths are projected to be significantly aggravated under climate change scenarios, particularly under the RCP8.5 scenario, with nearly twice the amount of excess heat-related deaths in the 2090s than under the RCP4.5 scenario. The RCP8.5 scenario assumes that the concentration of greenhouse gas will consistently increase in conjunction with high energy consumption, excessive land use, and high population growth. Considering the complexities of the challenges posed by future global warming, integrated multisectoral and multidisciplinary collaborations linking energy generation, transport, industry, and agriculture are needed for the development of policies to reduce man-made greenhouse gas emissions to slow the pace of global warming[34]. Urban areas are anticipated to have faster temperature increases in the future[2], and local governments can play a central role in adapting to and mitigating climate change hazards through the implementation of heat-health early warning systems and active heat response plans, the effective allocation of medical resources, and the construction of well-designed infrastructures, such as green spaces, green roofs, and wind corridors. Additionally, as future heat-related risk might be partly influenced by urbanization and population aging, it is recommended that health adaptation and mitigation policies and strategies consider these important factors for building a healthier and more resilient living environment to address the adverse effects of climate change[35]. Health professionals could play an important role in

communicating with local and national policymakers and the public about the health-related dangers of extreme weather and recommended active preventive measures. Furthermore, patients with cardiorespiratory diseases, those residing in the southern, eastern, central, and northern regions, females, the elderly, and the illiterate are expected to be particularly sensitive to future high temperatures. Tailored climate-health intervention strategies implemented by the local community and public health centers should be targeted to these susceptible populations. For individuals, awareness of climate change hazards needs to be enhanced, particularly for vulnerable populations; and active personal behaviors, such as reducing outdoor activity during the hottest hours of the day, using air conditioning and ventilation, wearing loose clothing and hats, and drinking sufficient fluids, are recommended during hot weather.

Limitations of our study should be noted. First, similar to data in previous studies on the health risk assessment of climate change[7,13,21], data on historical temperatures were collected from fixed weather monitoring stations rather than individual-level exposure, which may introduce measurement errors. However, these errors are likely to be randomly distributed. Second, the age-specific fertility rates vary between urban and rural areas[32]. The linkage between the age-specific population change ratio and urbanization in the future cannot be fully represented by current SSP scenarios. Our study may have obtained a conservative estimate of future heat-related excess mortality, and future studies are still required to appropriately assess the complex aspects of future population structural changes under SSP scenarios. Third, as low and high temperatures consistently presented different exposure–response associations[23,36,37], such as distinct lag patterns and harvesting effects, the present investigation only focused on the mortality risk assessment of future high temperatures. The lack of estimates of cold-related deaths may limit the assessment of net excess deaths due to future climate change. Findings as to whether the increased impact of high temperatures can be compensated by a reduction in the health burden of cold temperatures under climate change scenarios remain inconsistent. For instance, a recent multicountry study observed a reduced net effect of future low and high temperatures in northern Europe, East Asia, and Australia, but an increasing trend in regions with hotter climates, such as South America, Southern Europe, and Southeast Asia[13]. Furthermore, previous studies have warned that we should not simply use the current temperature-mortality relationship to project future cold-related excess deaths[38–40], and climate change is unlikely to dramatically reduce cold-related mortality in the future[39–41]. In addition to future temperature increases, temperature variability and the characteristics of extreme temperature events (i.e., duration, timing, and intensity of heatwaves and cold spells) may be altered by future climate change. Future research is needed to consider the health effects of such potential changes to provide a comprehensive picture of the impacts of global warming and climate change.

In conclusion, we find that future temperature increases will lead to significantly increased burdens of heat-related mortality in China, particularly under the RCP8.5 scenario. People living in southern, eastern, central, and northern China, subjects with cardiorespiratory diseases, females, the elderly, and people with lower educational levels will suffer more adverse effects from future high temperatures in China. Population aging will remarkably amplify heat-related deaths in the future.

## Methods

**Study area and data sources**. The Disease Surveillance Points system (DSPs) in China contained 161 surveillance points (each point represents a county or a district of a city) during 2007–2013, including 64 districts and 97 counties, covering 73 million people. The DSPs record all causes of death and population counts at the surveillance sites, which are nationally representative. A detailed description of the sampling strategy and the characteristics of the DSPs can be found in the study of Zhou et al.[42], in addition to the quality control processes and procedures for collecting data and coding the cause of death. To ensure the generalization of our results to the whole country, our study included all 161 districts/counties in the DSPs[43], and then they were further classified into seven regions, including 23 districts/counties in the north, 18 in the northeast, 20 in the northwest, 26 in the central, 34 in the east, 26 in the southwest and 14 in the south (Supplementary Fig. 1a).

**Historical mortality, weather, and socioeconomic data**. Causes of death were defined using the Tenth Revision of the International Classification of Diseases (ICD-10), including non-accidental death (ICD-10: A00-R99), cardiovascular disease (I00–I99), stroke (I60–I69), ischemic heart disease (I20–I25), respiratory ailments (J00–J99), and chronic obstructive pulmonary disease (J40–J47). In addition, we further divided the daily counts of non-accidental mortality by individual characteristics, including gender, age group (0–74 and 75+ years), and educational attainment (the illiterate and primary education or above).

Daily weather data on minimum, maximum and mean temperatures (°C), mean relative humidity (%), and mean wind speed (m/s) from 839 weather stations (Supplementary Fig. 1b) during 2007–2013 were collected from the China Meteorological Data Service Center (http://data.cma.cn/). All these weather data are strictly quality controlled by the China Meteorological Administration and the Resources and Environmental Science Data Center, Chinese Academy of Sciences. The daily meteorological records at 839 stations were interpolated separately into 1 km × 1 km gridded data using the inverse distance weighting interpolation technique in ArcGIS 10.5 (Environmental Systems Research Institute, Redlands, CA, USA). Then, the daily meteorological variable for each district/county was calculated by averaging the value of the grid cell within its boundary, which could dilute the potential bias from an individual station. The GDP per capita for each district or county was derived from the statistical yearbooks at the provincial or district/county levels.

The daily concentrations of fine particulate matter ($PM_{2.5}$) and ozone ($O_3$) at a horizontal resolution of 36 km × 36 km during 2011–2013 were predicted by the modified community multiscale air quality model. Detailed information on the methodology was provided in previous studies[44,45]. In brief, the model inputs included the meteorological parameters generated based on the weather research and forecasting model, the anthropogenic emissions generated from the multiresolution emission inventory, the biogenic emissions based on the model for emissions of gases and aerosols from nature, and biomass burning emissions based on the fire inventory. The evaluation of the model's prediction capability was performed using monitoring data from 422 sites in 60 large cities across China[44]. Concentrations of air pollution for each district or county were calculated by aggregating the values of the grid cells within its boundary[43].

**Future temperature data**. Projected daily minimum and maximum temperatures in each district or county during 1960–2099 were statistically downscaled by the weather generator-based statistical downscaling model (NWAI-WG)[46]. Specifically, we first extracted the monthly projected temperature in 28 GCMs from datasets of the 5th Coupled Model Intercomparison Project phase 5 (CMIP5) under RCP4.5 and RCP8.5 (Supplementary Table 2). The selection of these two RCP scenarios was motivated by previous studies[47–49]. Specifically, RCP4.5 assumes that global annual greenhouse gas emissions will stabilize by approximately 2040 and then decline, while the RCP8.5 scenario assumes a continuing rise in emissions in the absence of climate change mitigation policies. These two scenarios represent relatively better (RCP4.5) and worst (RCP8.5) perspectives for future emissions over this century. The GCM projected monthly values for a site were spatially downscaled from four nearby grids rather than from the nearest-grid GCM, using an inverse interpolation method. This approach can avoid multiple counties within one grid sharing the same projected value and can provide better representations of spatial changes in projected temperatures. The bias correction of GCM-projected monthly values and the historically observed temperatures were performed using an equidistant quantile method. Then, daily temperature measures were disaggregated through a modified weather generator for the daily weather variables[50]. The details of the downscaling procedure can be found in the previous study[46]. In addition, when the projected temperature series are applied to the temperature-mortality association estimated using observed temperatures, deviations within the data source may be nonnegligible. Therefore, we further corrected the projected daily temperature series using the bias-correction method by Hempel et al.[51] that preserves the long-term absolute temperature change. Finally, we calculated the average of daily minimum and maximum temperatures as the daily mean temperature to project heat-related excess mortality.

**Population projection**. The gridded population projections for China during 2010-2099 under five SSPs at a 1 km × 1 km resolution were extracted from the SSP spatial population scenario database (www.cgd.ucar.edu/iam/modeling/spatial-population-scenarios.html). The population projections for each district or county were computed by aggregating the populations of grid cells within its boundary. Considering the bias between the projected population and census data, we

calculated the district/county-specific correction factor by comparing the population for 2010 from the SSP dataset and the 2010 population census from the National Bureau of Statistics (http://www.stats.gov.cn/tjsj/pcsj/). The correction factor is assumed to be constant and is used to adjust the SSP-specific and district/county-specific projected population under five SSPs[52]. The present study considered six population scenarios in China: a no population change scenario assuming that the population structure remains constant from 2010 to 2099, and five SSPs. The SSPs illustrate a set of future pathways of socio-economic development in the absence of new climate policies and climate change mitigation through the end of this century, which are featured by a function of different levels of fertility, mortality, migration, and education for future population growth. A detailed description of the SSPs can be found in the previous studies[7,53].

To reveal the influence of population aging on the future heat-related burden, we estimated the district/county-specific future projected population sizes between 2010 and 2099 for two age groups (0–74 years and 75+ years) under five SSPs[27]. As data on future age structure are only available at the provincial and municipality levels, we used the composition ratios for these age groups during 2010–2099 at the provincial and municipality levels for the subordinated districts/counties. The projected population of different age groups is presented in Supplementary Fig. 6.

**Historical temperature–mortality relationship**. The historical association between temperature and mortality was estimated by a two-stage analysis. In the first stage, we used the quasi-Poisson regression with the distributed lag non-linear model (DLNM) to fit the district/county-specific relationship between temperature and mortality[54]. The district/county-specific model is as follows:

$$\mathrm{Log}\left[E\left(Y_{i,t}\right)\right] = \alpha + S(t; \beta) + \gamma \mathrm{Dow}_{i,t} + f(T_{\mathrm{obs}}; \theta) \tag{1}$$

where $Y_{i,t}$ is the observed number of deaths at surveillance point $i$ ($i = 1, 2, 3, \ldots, 161$) on day $t$ ($t = 1, 2, \ldots, 2557$); $\alpha$ is the model intercept; and $S(t; \beta)$ represents a natural cubic spline with seven degrees of freedom (df) per year to adjust for seasonality and the time trend of mortality. Day of the week was included as a categorical variable. These model specifications were motivated by previous studies[36,54–56]. Then we fitted a cross-basis term $f(T_{\mathrm{obs}}; \theta)$ generated by DLNM to capture the non-linear and delayed effects of temperature on mortality. Instead of the quadratic B-spline, a natural cubic spline was selected in the bidimensional cross-basis function, which allowed the log-linear extrapolation of the function beyond the boundaries of the observed temperature series, a step required to model the mortality risks of projected temperatures[13,57]. Specifically, we modeled the cross-basis function for the temperature-mortality dimension using a natural cubic spline with internal knots at the 10th, 75th, and 90th percentiles of the district/county-specific temperature distributions, and a natural cubic spline with three internal knots at equally spaced values of the lag dimension in the log scale. Since our previous studies indicated that the impact of high temperature on mortality was limited to 2 weeks after considering the potential harvesting effect[23,58], the maximum lag of 14 days was applied to capture the delayed effect of temperature. The cumulative temperature–mortality association in each district or county was represented using relative risk by comparing the risk at each temperature series to the minimum-mortality temperature ($T_{\mathrm{mm}}$), which is the temperature at which the mortality risk was the lowest[13].

In the second stage, a multivariate meta-analysis based on restricted maximum likelihood was performed to examine the pooled temperature–mortality relationship. To partly account for the spatial heterogeneity in the relationship, we included the district/county-specific average temperature, temperature range, GDP per capita, urban or rural community indicator, and region indicator in the model as meta-predictors. Then, the best linear unbiased prediction (BLUP) was used to predict the cumulative temperature–mortality relationship for each district or county. Details of this methodology have been described in previous studies[59,60]. The heterogeneity was tested by Cochran's Q method and extension of the $I^2$ statistic.

**Projecting future heat-related mortality**. Based on the nonlinear and lagged temperature-mortality relationship, we calculated the daily historical and future number of heat-attributable number deaths $D_{\mathrm{heat}}$ on any day with daily mean temperature above the $T_{\mathrm{mm}}$ as

$$D_{\mathrm{heat}} = D \cdot \left(1 - e^{-\left(f^*\left(T_{\mathrm{proj}}^*; \theta_b^*\right) - f^*\left(T_{\mathrm{mm}}^*; \theta_b^*\right)\right)}\right) \tag{2}$$

where $f^*$ and $\theta^*$ denote the overall cumulative temperature-mortality association derived from the bidimensional term. $T_{\mathrm{proj}}^*$ represents the projected temperature series. The number of death attributable to high temperatures was calculated by summing subsets of days with temperatures above $T_{\mathrm{mm}}$. Then, the attributable fraction of heat-related deaths was obtained by dividing the number of heat-attributable deaths by the total number of deaths. Detailed methodological steps can be found in a recently published hands-on tutorial[57].

Decadal heat-related excess mortality was separately estimated for each district/county and combinations of RCPs and GCMs. Subsequently, attributable fractions as GCM-ensemble means according to region, decade and RCP were further calculated using the corresponding total number of deaths as the denominator[13].

**Uncertainty analysis**. The uncertainties in the projection of future heat-related mortality are mainly from the temperature–mortality relationship, the variation in projected temperatures from different GCMs, and the population projections. The uncertainty in the projected populations was considered using five credible SSP scenarios. The uncertainties in sources of the baseline temperature–mortality association are represented by the variance of model coefficients $V(\theta_b^*)$ and the variation in future projected temperature series ($T_{\mathrm{proj}}^*$) among the 28 GCM models. These uncertainties were quantified by producing 1000 samples of reduced BLUP coefficients using Monte Carlo simulation, assuming that the estimated spline model coefficients followed a multivariate normal distribution, and then generating results for each GCM[13,57]. We reported the excess mortality as GCM-ensemble averages based on decades and RCP scenarios. The 95% empirical confidence intervals (eCI) were defined as the 2.5th and 97.5th percentiles of the empirical distribution across coefficient samples and GCMs.

Furthermore, the robustness of the modeling parameters was tested by altering the maximum lag for the temperature–lag–mortality relationship from 8 to 21 days and degrees of freedom for long-term and seasonal trends of mortality from 6 to 10 per year. As another sensitivity analysis, we separately included daily mean relative humidity and mean wind speed in the main model using a natural cubic spline with 3 degrees of freedom assessments. In addition, we adjusted for the 2-day average concentrations of $PM_{2.5}$ and $O_3$ by natural cubic spline with 3 degrees of freedom using data for 2011–2013, as air pollution data were only available for this period.

**Vulnerable subgroups and population aging**. To identify vulnerable subpopulations to future high temperatures, we separately conducted the above-mentioned analyses by cause of death, gender, age group, and educational attainment. We projected the attributable fraction and an attributable number of deaths due to future high temperatures in different subpopulations based on the corresponding pooled temperature–mortality relationship under the assumption of no changes in population structure.

In addition, to consider the influence of population change and aging on the future heat-related burden, we estimated the age-specific number of heat-attributable deaths by considering the age-specific baseline temperature–mortality association, and the age-specific population change ratio under five SSPs in the future relative to the 2010s[61,62].

All the data preparations and analyses were conducted through the R Foundation for Statistical Computing, version 3.5.1. The "dlnm" and "mvmeta" packages were separately applied to build the DLNM and to conduct the multivariate meta-analysis[59,63]. A two-tailed $P$ less than 0.05 was considered statistically significant for all statistical tests. Maps were generated using ArcGIS software, version 10.3.

**Reporting summary**. Further information on research design is available in the Nature Research Reporting Summary linked to this article.

## Data availability

The dataset generated and analyzed during this study is available (with some institutional limitations) from the corresponding authors upon reasonable request. The mortality data can be obtained from the Chinese Center for Disease Control and Prevention (China CDC) under the agreement to not engage in the unauthorized distribution of the raw data to a third party and to use the data for scientific research only (http://www.phsciencedata.cn/Share/en/index.jsp). Historical temperature data are available at the dataset of daily climate data from Chinese surface stations of the China Meteorological Data Service Center (http://data.cma.cn/). The NWAI-WG statistically downscaled CMIP5 GCM dataset for the observed climate sites from the China Meteorological Data Service Center are available at https://www.agrivy.com/. Gridded population projections of China for five SSPs are available at the Spatial Population Scenario database (www.cgd.ucar.edu/iam/modeling/spatial-population-scenarios.html). Air pollution data are available on request from Prof. Jianlin Hu from Nanjing University of Information Science and Technology (jianlinhu@nuist.edu.cn). The source data underlying Figs. 1–4 are provided as Supplementary Data files.

## Code availability

The code is available on request.

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

## Acknowledgements

The study was supported by the National Natural Science Foundation of China (No. 82003552), Guangdong Basic and Applied Basic Research Foundation (No. 2020A1515011161), Natural Science Foundation of Guangdong Province (No. 2018A030310655), the National Key Research and Development Program of China (No. 2018YFC0213602), the National Basic Research Program of China (973 Program) (No. 2012CB955504), Fundamental Research Funds for the Central Universities (No. 21618323) and Guangdong Province Science and Technology Department (2019B121202002, 2019B121205004, and 2019B110206002). Zhoupeng Ren was supported by the Strategic Priority Research Program of the Chinese Academy of Sciences (XDA20030302) and the National Natural Science Foundation of China (42071377, 41701460). Chun-Quan Ou was supported by the National Natural Science Foundation of China (81573249). Yuming Guo was supported by the Career Development Fellowship of the Australian National Health and Medical Research Council (APP1107107 and APP1163693). The funders played no role in determining the study design, data collection, or analysis methods employed, in our decision to publish, or in preparing the paper. We gratefully acknowledge Prof. Jianlin Hu from Nanjing University of Information Science and Technology for sharing the air pollution data and Dr. Bernie Dominiak of New South Wales Department of Primary Industries for his editing and review to improve the readability of the paper.

## Author contributions

J.Y. and Q.L. initiated the study. M.Z., Q.L., J.Y., P.Y., Z.P.R., and D.L.L. collected the data. J.Y cleaned the data. J.Y. performed the statistical analysis. J.Y. drafted the paper. M.L., Y.M.G., B.G.W., C.Q.O., S.L.T., M.S., H.W., C.L.Z, J.F.W., and Q.L. revised the paper. All authors read and approved the final paper.

## Competing interests

The authors declare no competing interests.

## Ethical approval

This study was approved by the Ethics Committee of the Chinese Center for Disease Control and Prevention (No. 201214). Data were analyzed at an aggregate level and no participants were contacted.
