## [Peer Review File · Nature Communications]

Reviewers' comments, initial round:

Reviewer #2 (Remarks to the Author):

The statistical analysis of this study was performed rigorously. I have two main concerns on the study being published in this journal:

(1) The historical temperature-mortality relationship was developed based on the mortality data covering a small portion of the Chinese population (73 million, or roughly 5 percent), but the authors are trying project the mortality burden for the entire nation

(2) Novelty of the study: This paper seems to use exactly the same method of an earlier global study (Gasparrini et al, 2017, Projections of temperature-related excess mortality under climate change scenarios. *The Lancet Planetary Health*, 1(9), e360-e367.) and applied it to a different dataset in China (mortality data in 161 Chinese communities, 2007-2013). Another recent study (Wang et al. 2019, Tens of thousands additional deaths annually in cities of China between 1.5° C and 2.0° C warming. *Nature communications*, 10(1), 1-7.) presents a better projection of heat-related mortality in China in my opinion, in terms of covering a much larger population and differentiating age groups in more details.

Other comments:

Line 67-69: "To date, the current investigations have...over a short time period": This statement was cited from a 2011 paper and is no longer valid. Refer to more recent studies, for example: Wang et al. 2019, Tens of thousands additional deaths annually in cities of China between 1.5° C and 2.0° C warming. *Nature communications*, 10(1), 1-7.

Li et al. 2018, Projecting future climate change impacts on heat-related mortality in large urban areas in China. *Environmental research*, 163, 171-185.

Line 106 and Figure 2: the historical data years (1971-2005) are not consistent with the years presented in the method section (Line 321, years are 1980-2099)

Line 111: "showing obvious geographical differences": Figure 2 does not show geographical variation.

Line 112-114: It is difficult to see from Table S3 the fastest changing region in temperature and the slowest changing region as well. Maybe include change in temperature over time in addition to the absolute temperature.

Line 157: Table S7 should be S9.

Line 159-161: the "first study" statement is untrue. For example two recent national studies I mentioned above. Wang et al (2019) covered 247.6 million population, Li et al (2018) covered 460 million population, whereas this study only covered 73 million.

Line 178-181: Reference#15 reported the estimated heat-related mortality rate in the past and future and the authors converted those into percentage changes. It would be easier for readers to understand if the original rates were cited.

Line 184-187: However, there were studies reporting the colder regions may be affected more than warmer regions. See for example: Schwartz et al (2015). Projections of temperature-attributable premature deaths in 209 US cities using a cluster-based Poisson approach. *Environmental Health*, 14(1), 85.

Line 271-273: I think the authors can actually analyze the cold-related mortality with their data.

Line 320-324: What is the spatial resolution of the projected weather data from the GCMs?

Line 328: RCP8.5 should be the "worst" scenario under current projection.

Line 335: be found IN previous study

Line 354-359: Using a single national population growth rate to project the population structure change for all 161 communities is way too simplified. This approach completely ignores possible population migration in the future.

Figure 1 on Page 18: I suggest showing data for the entire study period (2007-2013) instead of a single year of 2010

Figure 4 on Page 21: The Figure title is too long. Line 611-615 simply repeated the same information three times, and may be completely removed.

Finally, earlier studies suggested globally the effects of heat on mortality may be very different in urban versus rural areas. It is unclear the 161 communities in this study are urban or rural areas. And again, it may not be reasonable to consider the estimates for these communities as a national projection.

Reviewer #3 (Remarks to the Author):

The increase in surface temperature in China has been faster than the global rate and the population aging trend is significant in China. Due to these facts, it is extremely important to assess the adverse health effects of high temperature on health in China. There are a few studies focused on heat-related health impacts in China, but they often ignored the changing population structure and adaptation capacity. A previous research assessed the annual heat-related mortality in densely populated cities of China at 1.5 °C and 2.0 °C global warming. With the gridded population projections of China during 2010-2100 under five shared socioeconomic pathways (SSPs) at 1km×1km resolution, this study extended the previous work, considering the change of population structure. The cumulative exposure-response curves for different mortality categories, sex, age groups and educational attainments are valuable for climate adaptation.

Here are some questions and comments regarding to the data and results:

1. What is the definition of community in this paper? Is it a city or prefecture?
2. It is stated in the paper that the weather data were collected from one basic weather monitoring station in each community. How was the meteorological data matched to a community? As I know China has about 700 basic referencing station with systematically calibrated data. Not all weather monitoring stations have good data quality control. Due to the fast urbanization process in China, many weather monitoring stations have been relocated, which should be noticed in data preparation.
3. As air conditioning can greatly reduce the heat-related mortality, why don't use income in categorization? Income/family income might be a more powerful indicator than education attainments.
4. In the abstract: the decadal heat-related excess mortality increased from 1.1% (95%eCI: 0.2, 2.0) in the 2010s to 4.5% (95%eCI: 0.5, 8.6) in the 2090s under RCP8.5, with corresponding 46,389 (95%CI: 8,573, 82,410) and 189,416 (95%CI: 20,000, 359,568) excess deaths, assuming no population changes. The long-term trend of temperature-related health burden is crucial to climate adaptation. Due to the huge uncertainty in the projection, the values may not be very convincing. Why don't we consider 2030 and 2050? The trend of change may be more meaningful than exact values.
5. Considering population aging in the projection is an improvement of this work. The author may want to address more in the abstract.
6. Urbanization is an important driver of global warming, at the same time it greatly reshapes the spatial distribution and structure of population in a region. This can be another source of uncertainty in the projection.
7. The uncertainties in the projection of future heat-related mortality are mainly from the temperature-mortality relationship, the variation in the projected temperature, as well as the

population projection. The author may need to elaborate the uncertainty in population structure in discussion as well.

There are some typos and grammar mistakes in the paper. E.g. on page 12 "The "dlnm" and "mvmeta" packages was..." should be "The "dlnm" and "mvmeta" packages were..." On the same page, "Two-tailed P less than 0.05 were..." should be "Two-tailed P less than 0.05 was..." The author may want to carefully check the manuscript.

Reviewers' comments:

Reviewer #2 (Remarks to the Author):

The statistical analysis of this study was performed rigorously. I have two main concerns on the study being published in this journal:

Response: Thanks for the supportive comment.

(1) The historical temperature-mortality relationship was developed based on the mortality data covering a small portion of the Chinese population (73 million, or roughly 5 percent), but the authors are trying project the mortality burden for the entire nation

Response: The daily mortality data in 161 communities (64 city districts in urban areas and 97 counties in rural areas) were collected from the nationwide Disease Surveillance Points system (DSPs), which is under management and operation by the Chinese Center for Disease Control and Prevention (China CDC). This system has created a good representative database on disease surveillance and mortality in China¹, in terms of geographical and population distribution of DSPs (Figure A).

Figure A. Distribution of Disease Surveillance Points system (161 sites) in China.

The DSP system was established using the multi-stage random sampling strategy and is representative of national status of disease surveillance and mortality¹, which involves 161 points (64 city districts and 97 rural counties) and covers 73 million population. The data from DSPs have been extensively used in the regular health reports by the government, and burden of disease studies by scientists. For example, Zhou et al. (2016) estimated 240 cause-specific mortality rates during 1990-2013 at both provincial and national levels in the mainland China (Lancet 2016; 387: 251–72)²; Wang and colleagues assessed the under-5 mortality rate for 31 provinces between 1970 and 2013 in China (Lancet 2016; 387: 273–83)³. Therefore, we are confident about the national representativeness of the mortality data during 2007-2013 in this study.

(2) Novelty of the study: This paper seems to use exactly the same method of an earlier global study (Gasparrini et al, 2017, Projections of temperature-related excess mortality under climate change scenarios. The Lancet Planetary Health, 1(9), e360-e367.) and applied it to a different dataset in China (mortality data in 161 Chinese communities, 2007-2013). Another recent study (Wang et al. 2019, Tens of thousands additional deaths annually in cities of China between 1.5° C and 2.0° C warming. Nature communications, 10(1), 1-7.) presents a better projection of heat-related mortality in China in my opinion, in terms of covering a much larger population and differentiating age groups in more details.

Response: As mentioned above, one of the most important novelties of this study is that the daily mortality in this study were collected from 161 DSPs (64 city districts and 97 rural counties), which have better national representativeness than any previous reports. For instance, Wang et al. (2019)⁶ used mortality data from 12 Chinese capital cities to establish the baseline temperature-mortality association, and then projected the future heat-related excess death for 27 metropolises. Therefore, their results may have large uncertainty on the projection of future heat-related mortality as they established baseline temperature-mortality associations based on limited number of cities.

In addition to the data source, our study provided more comprehensive assessment of vulnerability to heat impacts, compared with previous reports. We projected the future heat-related excess mortality by individual characteristics, including causes of death (non-accidental mortality, cardiovascular disease, ischemic heart disease, stroke, respiratory disease and chronic obstructive pulmonary disease), gender, age group (0-74 and 75+ years), and educational attainments (the illiterate, and primary education or above). Additionally, we assessed the future heat-related excess mortality by region in China, which hasn't been looked at by any previous study. In addition, we have assessed the influence of population change and aging on future heat-related burden. Therefore, this study greatly extended the previous work, and can reduce the uncertainties in the projected future heat-related burden.

Other comments:

Line 67-69: "To date, the current investigations have...over a short time period": This statement was cited from a 2011 paper and is no longer valid. Refer to more recent studies, for example:

Wang et al. 2019, Tens of thousands additional deaths annually in cities of China between 1.5° C and 2.0° C warming. *Nature communications*, 10(1), 1-7.

Li et al. 2018, Projecting future climate change impacts on heat-related mortality in large urban areas in China. *Environmental research*, 163, 171-185.

Response: We have revised these sentences and updated the references as suggested.

Line 106 and Figure 2: the historical data years (1971-2005) are not consistent with the years presented in the method section (Line 321, years are 1980-2099)

Response: Sorry for this mistake. The period of the historical data should be 1971-2005. We have corrected it in the methodology section (Line 384).

Line 111: "showing obvious geographical differences": Figure 2 does not show

geographical variation.

**Response: We have removed “showing obvious geographical differences”.
Thanks.**

Line 112-114: It is difficult to see from Table S3 the fastest changing region in temperature and the slowest changing region as well. Maybe include change in temperature over time in addition to the absolute temperature.

Response: We have reconstructed this table and added the change in temperature during 2030s, 2050s and 2090s relative to the baseline of 2010s as suggested.

Line 157: Table S7 should be S9.

Response: Done. Thanks.

Line 159-161: the "first study" statement is untrue. For example two recent national studies I mentioned above. Wang et al (2019) covered 247.6 million population, Li et al (2018) covered 460 million population, whereas this study only covered 73 million.

Response: We have changed this phrase as “one of few studies” (Lines 182-183).

Line 178-181: Reference#15 reported the estimated heat-related mortality rate in the past and future and the authors converted those into percentage changes. It would be easier for readers to understand if the original rates were cited.

Response: Thank you for this useful comment. We have added the heat-related mortality rate in the past and future in this sentence as “Petkova and colleagues reported comparable estimates of 171.1-286.5% and 255.6-410.8% increases in heat-related deaths in decade of 2050s under RCP4.5 and RCP8.5 (heat-related mortality rates: 8.8-17.1 per 100,000 and 11.7-16.0 per 100,000) in three major USA cities, compared to the baseline of 1971 to 2000 (2.9-4.5 per 100,000), with the corresponding estimates of 237.8-362.2% and 537.8-827.0% in the 2080s (10.5-17.1 per 100,000 and 19.3-34.3 per 100,000)” (Lines 201-204).

Line 184-187: However, there were studies reporting the colder regions may be affected more than warmer regions. See for example: Schwartz et al (2015). Projections of temperature-attributable premature deaths in 209 US cities using a cluster-based Poisson approach. *Environmental Health*, 14(1), 85.

Response: Thank you for this comment. We have included the mentioned study in this sentence, and explained that this discrepancy may be partly related to the spatial heterogeneity in the vulnerability of local residents to heat exposure, and differences in the study design and GCM models used [only two GCM models (GFDL-CM3 and MIROC5) were used in the study of Schwartz et al (2015)⁸] (Lines 212-214).

Line 271-273: I think the authors can actually analyze the cold-related mortality with their data.

Response: We analyzed the baseline temperature-mortality associations using both cold and hot temperatures, but only projected the death burden of future high temperature for the following reasons: Firstly, heat-related impacts are a major public health concern as climate change proceeds. Secondly, the high and low temperatures present different exposure-response associations, including the lag pattern and harvesting effect. The health effect of high temperature was immediate and limited to several days, but often followed with harvesting effect. While the health effect of low temperature is less direct and could last for several weeks. Therefore, Lo and colleagues⁹ suggested that the heat- and cold-related mortality should be studied separately. Thirdly, some scientists warned that it is a great challenge to apply the current temperature-mortality relationship to project the future cold-related excess deaths when the seasonal confounding (ie, influenza) was not adequately adjusted for¹⁰⁻¹².

Line 320-324: What is the spatial resolution of the projected weather data from the GCMs?

Response: The spatial resolution of these GCMs simulations in this study ranged

from 0.7° (lat.) \times 0.8° (lon.) to $3.7^{\circ}\times 3.78^{\circ}$. We have added the resolution of each GCM to Supplementary Table 1. The downscaling model involves a spatial downscaling which interpolates four nearby grids rather than the nearest-grid GCM. This approach can avoid multiple counties within one grid sharing the same projected values. The downscaling method, Nwai-WG, used the four nearby grids with weighed interpolation based on their distances from each center of the grids¹³. This method can provide better representations of spatial changes¹³.

For clarification, we modified the sentences “Projected daily minimal and maximal temperatures in each community during 1980-2099 were obtained.” and “These two scenarios represent relatively “better case” and “worse case” perspectives for future emissions of over this century, respectively. An inverse interpolation method then was applied to downscale the data to the specific sites.” To “Projected daily minimal and maximal temperatures in each community during 1971-2099 were statistically downscaled by the weather-generator based statistical downscaling model (Nwai-WG)¹³” And “These two scenarios represent relatively “better case” (RCP4.5) and “worst case” (RCP 8.5) perspectives for future emissions of over this century. The GCM projected monthly values for a site were spatially downscaled from four nearby grids rather than the nearest-grid GCM, using an inverse interpolation method. This approach can avoid multiple counties within one grid share the same projected value and can provide better representations of spatial changes in projected temperatures.”, respectively.

Line 328: RCP8.5 should be the "worst" scenario under current projection.

Response: Done (Line 393).

Line 335: be found IN previous study

Response: Done (Line 401).

Line 354-359: Using a single national population growth rate to project the population structure change for all 161 communities is way too simplified. This approach completely ignores possible population migration in the future.

Response: In the revision, we collected the recent public open data¹⁴ on future demographic population under five shared socioeconomic pathways (SSPs) at the provincial and municipality level (27 provinces and 4 municipalities). These are the finest data available on demographic population projections under five SSPs in China. Then, we used the composition ratios for two age groups (i.e. 0-74 years and 75+ years) during 2010-2100 at provincial and municipality level for the subordinated communities. As the population structure and socioeconomic status are quite similar within the same province or municipality, these new data have taken into account the possible migration in the future (Lines 427-430).

Figure 1 on Page 18: I suggest showing data for the entire study period (2007-2013) instead of a single year of 2010

Response: We have updated this figure using the community-specific annual mean temperature and mortality rate based on data for the entire study period (2007-2013).

Figure 4 on Page 21: The Figure title is too long. Line 611-615 simply repeated the same information three times, and may be completely removed.

Response: The repeated information has been removed in the revision, and this Figure title was changed as “Figure 4. The heat-related attributable number of deaths in the 2030s (A-C), 2050s (D-F) and 2090s (G-I) for different age groups under six population scenarios (no change, SSP1, SSP2, SSP3, SSP4 and SSP5) in China.”.

Finally, earlier studies suggested globally the effects of heat on mortality may be very different in urban versus rural areas. It is unclear the 161 communities in this study are urban or rural areas. And again, it may not be reasonable to consider the estimates

for these communities as a national projection.

Response: According to the 6th national census of population (2010 census), there were 665.6 million urban dwellers and 674.1 million rural residents in China (<http://www.stats.gov.cn/english/Statisticaldata/CensusData/>). Therefore, to reasonably produce a national estimation, it is essential to include the data from both urban and rural population. However, previous studies only used the data from cities (urban population)^{6,15}. By stark contrast, the mortality data used in this study were collected from 161 DSPs (64 city districts and 97 rural counties), which provided good representativeness of mortality in China.

And to account for the spatial heterogeneity in the relationship, we have included the community-specific average temperature, temperature range, GDP per capita, indicator for urban or rural community, and indicator for regions in the multivariate meta-analysis as meta-predictors, a method commonly used in many previous studies^{4,16,17}.

Reviewer #3 (Remarks to the Author):

The increase in surface temperature in China has been faster than the global rate and the population aging trend is significant in China. Due to these facts, it is extremely important to assess the adverse health effects of high temperature on health in China. There are a few studies focused on heat-related health impacts in China, but they often ignored the changing population structure and adaptation capacity. A previous research assessed the annual heat-related mortality in densely populated cities of China at 1.5 °C and 2.0 °C global warming. With the gridded population projections of China during 2010-2100 under five shared socioeconomic pathways (SSPs) at 1km×1km resolution, this study extended the previous work, considering the change of population structure. The cumulative exposure-response curves for different mortality categories, sex, age groups and educational attainments are valuable for climate adaptation.

Response: We thank the reviewer for the constructive critiques and comments.

Here are some questions and comments regarding to the data and results:

1. What is the definition of community in this paper? Is it a city or prefecture?

Response: The community refers to a district of a city in an urban area or a county of a city in a rural area. We have added the short definition of community in the methodology section (Lines 354-355), and also provided a table on the detailed information of each community in the supplemental material (Supplementary Table 1), such as the community code, the region information, population at different age, annual mean temperature and GDP per capita. This table may help the reader understand our data more easily.

2. It is stated in the paper that the weather data were collected from one basic weather monitoring station in each community. How was the meteorological data matched to a community? As I know China has about 700 basic referencing station with systematically calibrated data. Not all weather monitoring stations have good data

quality control. Due to the fast urbanization process in China, many weather monitoring stations have been relocated, which should be noticed in data preparation.

Response: In the revision, we collected all publicly open weather data from 839 weather stations (all stations) during 2007-2013 from the China Meteorological Data Service Center (<http://data.cma.cn/>). The distribution of the 839 weather stations has been added in Supplemental Fig. 1. Then, the meteorological records at 839 stations were interpolated separately for each day into 1km×1km gridded data using Inverse Distance Weighting interpolation technique in ArcGIS 10.5 (Environmental Systems Research Institute, Redlands, California, USA). The daily mean temperature for each community was calculated by averaging the value of grid cell within the community boundary, which may dilute the potential bias from an individual station (Lines 371-380). With these updated temperature data, we have reanalyzed all the data, but the new results are similar as original results.

3. As air conditioning can greatly reduce the heat-related mortality, why don't use income in categorization? Income/family income might be a more powerful indicator than education attainments.

Response: We agree that income/family income may be a more powerful indicator for the influence of air conditioning on heat-related mortality impact. However, DSPs only collect information of individual death, such as the date of death, birth of death, gender and educational level, and do not have the income information. In addition, average income of each community was also unavailable during 2007-2013. Therefore, we cannot perform the stratification analysis by the income. To address this issue, we collected the data on gross domestic product (GDP) per capita for each community in 2010 from the statistic yearbooks at the provincial or community levels, which could be viewed as an important economic performance and is closely correlated with the average income¹⁸. The results of the stratification analysis by the level of GDP per capita showed relatively higher heat-related excess mortality in communities with low

GDP per capita during 2010s, while the projected slope was steeper in communities with high GDP per capita in future decades (Lines 149-152; Supplementary Table 7). This interesting finding could support the hypothesis that air conditioning can reduce the health effect of high temperature at the baseline period, but urbanization in conjunction with heat island effect may aggravate the future heat effect in urban areas. We have also discussed this issue in the revision (Lines 218-222).

4. In the abstract: the decadal heat-related excess mortality increased from 1.1% (95%eCI: 0.2, 2.0) in the 2010s to 4.5% (95%eCI: 0.5, 8.6) in the 2090s under RCP8.5, with corresponding 46,389 (95%CI: 8,573, 82,410) and 189,416 (95%CI: 20,000, 359,568) excess deaths, assuming no population changes. The long-term trend of temperature-related health burden is crucial to climate adaptation. Due to the huge uncertainty in the projection, the values may not be very convincing. Why don't we consider 2030 and 2050? The trend of change may be more meaningful than exact values.

Response: It would be nice to provide the projected estimates in 2030s, 2050s and 2090s in the abstract, which represent the short-, median- and long-term scenario, respectively. However, according to the guideline of this journal, no more than 150 words are allowed in the abstract. Thus, we could only provide the main estimates of heat-related excess mortality (%) in 2030s (short-term) and 2090s (long-term). Furthermore, we have replaced the attributable number of heat-related mortality with the relative change (%) of excess mortality in future decades relative to the baseline period of 2010s as suggested (Lines 50-52).

5. Considering population aging in the projection is an improvement of this work. The author may want to address more in the abstract.

Response: We agree that it is worth adding more results of the influence of population aging on the future heat-related excess mortality. However, as the limited number of words is allowed for the abstract, we could just add the

information on the extent of the future heat-related excess mortality that would be increased by the population aging under SSPs compared to the population no change scenario in the abstract (Lines 55-57).

6. Urbanization is an important driver of global warming, at the same time it greatly reshapes the spatial distribution and structure of population in a region. This can be another source of uncertainty in the projection.

Response: It is important to acknowledge the uncertainty in projections of future temperature-related health risk, including the urbanization. Though we have exerted all our efforts to consider the uncertainty through using the most up-to-date methodology to assess the mortality burden of future high temperature using different climate models, emission scenarios, and future aging population under five SSPs, there is still uncertainty from urbanization, population adaptation and acclimatization, and population structure changes which should be noticed. Therefore, we have added a paragraph to carefully discuss this issue in the revised manuscript (Lines 270-293).

7. The uncertainties in the projection of future heat-related mortality are mainly from the temperature-mortality relationship, the variation in the projected temperature, as well as the population projection. The author may need to elaborate the uncertainty in population structure in discussion as well.

Response: Please refer to our response to the last comment.

There are some typos and grammar mistakes in the paper. E.g. on page 12 “The “dlnm” and “mvmeta” packages was...” should be “The “dlnm” and “mvmeta” packages were...” On the same page, “Two-tailed P less than 0.05 were...” should be “Two-tailed P less than 0.05 was...” The author may want to carefully check the manuscript.

Response: We have corrected these typos and checked the manuscript carefully.

References

1. Zhou, M.G., Jiang, Y., Huang, Z.J. & Wu, F. Adjustment and representativeness evaluation of national disease surveillance points system. *Disease Surveillance* **13**, 6295-6378 (2010).
2. Zhou, M., *et al.* Cause-specific mortality for 240 causes in China during 1990–2013: a systematic subnational analysis for the Global Burden of Disease Study 2013. *The Lancet* **387**, 251-272 (2016).
3. Wang, Y., *et al.* Under-5 mortality in 2851 Chinese counties, 1996–2012: a subnational assessment of achieving MDG 4 goals in China. *The Lancet* **387**, 273-283 (2016).
4. Gasparrini, A., *et al.* Projections of temperature-related excess mortality under climate change scenarios. *The Lancet Planetary Health* **1**, e360-e367 (2017).
5. Li, Y., Ren, T., Kinney, P.L., Joyner, A. & Zhang, W. Projecting future climate change impacts on heat-related mortality in large urban areas in China. *Environmental research* **163**, 171-185 (2018).
6. Wang, Y., *et al.* Tens of thousands additional deaths annually in cities of China between 1.5° C and 2.0° C warming. *Nature communications* **10**, 1-7 (2019).
7. Sanderson, M., Arbuthnott, K., Kovats, S., Hajat, S. & Falloon, P. The use of climate information to estimate future mortality from high ambient temperature: a systematic literature review. *PloS one* **12**(2017).
8. Schwartz, J.D., *et al.* Projections of temperature-attributable premature deaths in 209 US cities using a cluster-based Poisson approach. *Environmental Health* **14**, 85 (2015).
9. Lo, Y.E., *et al.* Increasing mitigation ambition to meet the Paris Agreement's temperature goal avoids substantial heat-related mortality in US cities. *Science advances* **5**, eaau4373 (2019).
10. Honda, Y., *et al.* Heat-related mortality risk model for climate change impact projection. *Environmental health and preventive medicine* **19**, 56 (2014).
11. Kinney, P.L., *et al.* Winter season mortality: will climate warming bring benefits? *Environmental Research Letters* **10**, 064016 (2015).
12. Ebi, K.L. & Mills, D. Winter mortality in a warming climate: a reassessment. *Wiley Interdisciplinary Reviews: Climate Change* **4**, 203-212 (2013).
13. Liu, D.L. & Zuo, H. Statistical downscaling of daily climate variables for climate change impact assessment over New South Wales, Australia. *Climatic change* **115**, 629-666 (2012).
14. Jiang, T., *et al.* National and provincial population projected to 2100 under the shared socioeconomic pathways in China. *Clim. Chang. Res* **13**, 128-137 (2017).
15. Li, T., *et al.* Heat-related mortality projections for cardiovascular and respiratory disease under the changing climate in Beijing, China. *Scientific Reports* **5**, 11441 (2015).
16. Gasparrini, A., *et al.* Mortality risk attributable to high and low ambient temperature: a multicountry observational study. *The Lancet* **386**, 369-375 (2015).
17. Onozuka, D., Gasparrini, A., Sera, F., Hashizume, M. & Honda, Y. Modeling future projections of temperature-related excess morbidity due to infectious gastroenteritis under climate change conditions in Japan. *Environmental health perspectives* **127**, 077006 (2019).
18. Diacon, P.-E. & Maha, L.-G. The relationship between income, consumption and GDP: A time series, cross-country analysis. *Procedia Economics and Finance* **23**, 1535-1543 (2015).

Reviewers' comments, second round:

Reviewer #2 (Remarks to the Author):

A main concern I have for the paper is the possible confounding effect of air pollution on daily mortality was not investigated based on the current statistical model. Air pollution levels remained high across China during the study period (2007-2013), and thus numerous premature deaths could be attributed to air pollution, but not high temperature (for instance, see: Yin, P., Chen, R., Wang, L., Meng, X., Liu, C., Niu, Y., ... & You, J. (2017). Ambient ozone pollution and daily mortality: a nationwide study in 272 Chinese cities. *Environmental health perspectives*, 125(11), 117006.). Particularly, ground-level ozone is highly correlated with temperature. The data used in this paper shows the mortality rates in northwest were much lower than those in the east. I think air pollution may play a role for this discrepancy. I suggest the authors test the effects of air pollution (PM2.5 and ground-level ozone) in their model and report the results in the article.

Reviewer #3 (Remarks to the Author):

The authors have addressed most of my questions well. I still have several questions for further clarification.

1. In China, there are about 100,000 communities. The sampled 161 communities account for 1.6% of the communities. The total population of China is about 1.4 billion, and the total population of these 161 communities is 73 million, which is about .5% of the total population of China. These two rates are somehow unmatched. I know that some prestigious academic journals have published several papers using the same set of data. After 4 years, do we have any updated data? If not, we can specify that the current study is based on the climate adaptation capacity in 2010s.

2. In recent 10 years, China has experienced a steady high speed of urbanization. The urban population increased from 45% in 2007 to more than 60% in 2019. This indicates a dramatic change in demographic features and adaptation capacity. On the other hand, urbanization has profound relationship with population aging. The age-specific fertility rate in rural and urban area has a significant difference in China (Luo et al. 2020, *The Lancet Global Health*). The linkage of age-specific population change ratio with urbanization in future remains unclear (cannot be fully represented by SSPs). In this sense, the estimated vulnerability to future heat-related excess mortality is less robust.

3. Line 214-222: what is the hypothesis? Is it "The rapid urbanization in conjunction with poor air quality and heat island effect may aggravate the heat effect in these highly developed regions in China"? The author should be cautious when using the word "confirmed" here. We don't have any causality analysis here.

4. Supplementary Table 7 : the projected slope was steeper in communities with high GDP per capita : this is an interesting finding. Is there any potential reason for that? What's the implication in policy-making?

5. Line 282: However, Baccini et al. (2011)¹⁴ argued: the reference style is inconsistent.

Figure 1 : the unit in Fig. 1 need to be consistent. The distance is in mile and the temperature is in Celsius degree. Also in Figure 1, the expression of one million is: 1,000,000 in the legend.

Reviewer #2 (Remarks to the Author):

A main concern I have for the paper is the possible confounding effect of air pollution on daily mortality was not investigated based on the current statistical model. Air pollution levels remained high across China during the study period (2007-2013), and thus numerous premature deaths could be attributed to air pollution, but not high temperature (for instance, see: Yin, P., Chen, R., Wang, L., Meng, X., Liu, C., Niu, Y., ... & You, J. (2017). Ambient ozone pollution and daily mortality: a nationwide study in 272 Chinese cities. *Environmental health perspectives*, 125(11), 117006.). Particularly, ground-level ozone is highly correlated with temperature. The data used in this paper shows the mortality rates in northwest were much lower than those in the east. I think air pollution may play a role for this discrepancy. I suggest the authors test the effects of air pollution (PM_{2.5} and ground-level ozone) in their model and report the results in the article.

Response: We thank the reviewer for this comment. Initially we did not examine the effect of air pollution because previous evidence has consistently showed stable effect estimates of temperature with and without the adjustment of air pollution¹⁻³. For example, in our previous nationwide study published in BMJ (the co-first author, Prof. Maigeng Zhou was the co-corresponding author of this paper)¹, the attributable fractions of non-accidental mortality due to ambient temperature were 13.14% (95%CI: 11.90-14.15%) and 13.28% (95%CI: 12.07-14.30%) before and after adjusting for fine particulate matter and ozone.

But to be prudent, as suggested by the reviewer, we have controlled for the two-day average concentrations of fine particulate matter and ozone using natural cubic spline with three degree of freedom as a sensitivity analysis in the revised manuscript (Lines 383-393, Pages 11-12; Lines 507-509, Page 15). The estimates did not materially change after adjusting for the exposure of air pollutants. For instance, under RCP8.5, the heat-related excess mortality were 1.9% (95%eCI: 0.2-3.3%) and 5.5% (0.5-9.9%) in the 2010s and the 2090s

without adjustment of air pollution, and 1.5% (0.4-2.6%) and 5.4% (0.9-9.8%) in the 2010s and the 2090s after adjusting for air pollution (Supplementary Table 10).

Reviewer #3 (Remarks to the Author):

The authors have addressed most of my questions well. I still have several questions for further clarification.

1. In China, there are about 100,000 communities. The sampled 161 communities account for 1.6% of the communities. The total population of China is about 1.4 billion, and the total population of these 161 communities is 73 million, which is about .5% of the total population of China. These two rates are somehow unmatched. I know that some prestigious academic journals have published several papers using the same set of data. After 4 years, do we have any updated data? If not, we can specify that the current study is based on the climate adaptation capacity in 2010s.

Response: Sorry for the confusions caused. We would like to clarify that the number of communities in China is 2851 (853 counties and 1998 districts) (China Statistical Yearbook 2019), not 100,000 as the reviewer used in his/her calculation. Our 161 communities accounted for 5.6% of all the communities ($161/2851 \times 100\%$); and the total population of these 161 communities accounted for 5.2% of all the populations in China ($73 \text{ million} / 1400 \text{ million} \times 100\%$). Therefore, these two rates are matched.

As to the data, we could not obtain any updated daily mortality data. And we have specified that the current study is based on the adaptative capacity in 2010s in the revised manuscript as suggested (Lines 49-50, Page 3; Lines 270-271, Page 8).

2. In recent 10 years, China has experienced a steady high speed of urbanization. The urban population increased from 45% in 2007 to more than 60% in 2019. This indicates a dramatic change in demographic features and adaptation capacity. On the other hand, urbanization has profound relationship with population aging. The age-specific fertility rate in rural and urban area has a significant difference in China

(Luo et al. 2020, The Lancet Global Health). The linkage of age-specific population change ratio with urbanization in future remains unclear (cannot be fully represented by SSPs). In this sense, the estimated vulnerability to future heat-related excess mortality is less robust.

Response: We thank the reviewer for this helpful comment. We assessed the population projection under five SSPs, which were featured by the different levels of fertility, mortality, migration and education, providing the most reliable strategy so far for projecting future population. As urbanization associated with population aging could increase the population vulnerability to heat exposure, we may have obtained a conservative estimate on the future heat-related excess mortality. This finding highlights the seriousness of climate change-related public health challenge, which could potentially be further exacerbated by population aging and urbanization.

In the revised manuscript, we have added the trend of urbanization in China in the discussion section (Lines 280-282, Page 9), carefully discussed the limitation, and underscored that future study is still warranted to appropriately assess more complex aspects of future population change under SSPs scenarios (Lines 322-327, Page 10).

3.Line 214-222: what is the hypothesis? Is it “The rapid urbanization in conjunction with poor air quality and heat island effect may aggravate the heat effect in these highly developed regions in China”? The author should be cautious when using the word “confirmed” here. We don’t have any causality analysis here.

Response: Yes, the hypothesis here is that urbanization and heat island effect is associated with a strong temperature increase and may aggravate the health risk of heat exposure in highly developed regions. The word “confirmed” was removed from our paper, and to avoid confusion caused, we have also re-organized these sentences in the revised manuscript (Lines 209-218, Page 7).

4. Supplementary Table 7: the projected slope was steeper in communities with high GDP per capita: this is an interesting finding. Is there any potential reason for that? What's the implication in policy-making?

Response: As the level of GDP per capita is highly correlated with the level of urbanization^{4,5}, we speculated that steeper slope of future heat effect in communities with high GDP per capita may be explained by the urbanization and other climate change vulnerability emerging in the urban areas, such as high population density and non-existent or poorly-established heat response plan for more frequency of future extreme heat. As these highly developed areas are anticipated to have faster temperature rise⁶ and steeper slope of heat-related risk in future, the local governments can play a central role in adapting and mitigating climate change hazards through the implementation of heat-health early warning systems and active heat response plans, the effective allocation of medical resources and the construction of well-designed infrastructures, such as green space, green roof and wind corridors. We have added the discussion related to the reasons and implications of this finding. (Lines 299-315, Pages 9-10).

5. Line 282: However, Baccini et al. (2011)¹⁴ argued: the reference style is inconsistent.

Response: Done. Thank you.

Figure 1: the unit in Fig. 1 need to be consistent. The distance is in mile and the temperature is in Celsius degree. Also in Figure 1, the expression of one million is: 1,000,000 in the legend.

Response: We thank the reviewer for this suggestion. The unit of mortality rates in the legend has been changed to “/1,000,000”. And the unit of distance was revised as kilometer, which is more common for the readers.

References

1. Chen, R., *et al.* Association between ambient temperature and mortality risk and burden: time series study in 272 main Chinese cities. *bmj* **363**, k4306 (2018).
2. Guo, Y., *et al.* Global variation in the effects of ambient temperature on mortality: a systematic evaluation. *Epidemiology (Cambridge, Mass.)* **25**, 781 (2014).
3. Yang, J., Ou, C.Q., Ding, Y., Zhou, Y.X. & Chen, P.Y. Daily temperature and mortality: a study of distributed lag non-linear effect and effect modification in Guangzhou. *Environ Health* **11**, 63 (2012).
4. Henderson, V. The urbanization process and economic growth: The so-what question. *Journal of Economic growth* **8**, 47-71 (2003).
5. Chen, M., Zhang, H., Liu, W. & Zhang, W. The global pattern of urbanization and economic growth: evidence from the last three decades. *PloS one* **9**, e103799-e103799 (2014).
6. IPCC. Summary for Policymakers. In: Climate Change 2013: The Physical Science Basis Contribution of Working Group I to the Fifth Assessment Report of the Intergovernmental Panel on Climate Change (ed. Stocker, T.F., D. Qin, G.-K. Plattner, M. Tignor, S.K. Allen, J. Boschung, A. Nauels, Y. Xia, V.Bex and P.M Midgley (eds.)) (Cambridge University Press, Cambridge, United Kingdom and New York, NY, USA, 2013).

Reviewers' comments, third round:

Reviewer #2 (Remarks to the Author):

I think the authors have addressed all my concerns. The manuscript is now in a good shape.

Reviewer #3 (Remarks to the Author):

Thanks for authors' hard work. Most of my concerns have been well addressed. I still have two more questions need some clarifications or revisions.

1. In the response to my comment about the community data, the authors claimed that the "community" in this research refers to "counties and districts". These are two different concepts in China. The "community" is generally a much smaller social and spatial scale unit than "counties and districts". The features of these two groups can be very different, especially in population structure. It is misleading to use "community" in this paper. I suggest the author clarify this term in the manuscript.

2. In addition to absolute temperature, other meteorological factors such as wind speed and relative humidity (Li and Chan, 2000) appear to have a notable effect on the level of heat-related death. Besides, hot days occurring early in the summer have been shown to be associated with greater impacts on mortality in the same population than later of comparable or higher temperatures (Hajat et al., 2002; Kinney et al., 2008; Anderson and Bell, 2011). I understand that estimate future mortality is difficult. Do we have any preliminary study to prove that the factors considered in this study are dominate in predicting future heat-related excess mortality in China?

REFERENCES

- Li, P.W. and S.T. Chan, 2000: Application of a weather stress index for alerting the public to stressful weather in Hong Kong. *Meteor. Appl.*, 7: 369–375.
- Hajat, S., R.S. Kovats, R.W. Atkinson and A. Haines, 2002: Impact of hot temperature on death in London: a time series approach. *J. Epidemiology and Community Health*, 56(5): 367–372.
- Kinney, P.L., M.S. O'Neill, M.L. Bell and J. Schwartz, 2008: Approaches for estimating effects of climate change on heat-related deaths: challenges and opportunities. *Environ. Sci. Pol.*, 11: 87–96.
- Anderson, G.B. and M.L. Bell, 2011: Heat waves in the United States: mortality risk during heat waves and effect modification by heat wave characteristics in 43 US communities. *Environ. Health Perspect.*, 119(2): 210–8.

Reviewer #2 (Remarks to the Author):

I think the authors have addressed all my concerns. The manuscript is now in a good shape.

Response: We appreciate the reviewer's supportive comments.

Reviewer #3 (Remarks to the Author):

Thanks for authors' hard work. Most of my concerns have been well addressed. I still have two more questions need some clarifications or revisions.

Response: We appreciate the reviewer's positive comments, and the suggestion for further clarifications.

1. In the response to my comment about the community data, the authors claimed that the "community" in this research refers to "counties and districts". These are two different concepts in China. The "community" is generally a much smaller social and spatial scale unit than "counties and districts". The features of these two groups can be very different, especially in population structure. It is misleading to use "community" in this paper. I suggest the author clarify this term in the manuscript.

Response: Thanks for pointing out. To avoid the confusion, we have changed the term "community" into "district/county" in the revised manuscript.

2. In addition to absolute temperature, other meteorological factors such as wind speed and relative humidity (Li and Chan, 2000) appear to have a notable effect on the level of heat-related death. Besides, hot days occurring early in the summer have been shown to be associated with greater impacts on mortality in the same population than later of comparable or higher temperatures (Hajat et al., 2002; Kinney et al., 2008; Anderson and Bell, 2011). I understand that estimate future mortality is difficult. Do we have any preliminary study to prove that the factors considered in this study are dominate in predicting future heat-related excess mortality in China?

REFERENCES

Li, P.W. and S.T. Chan, 2000: Application of a weather stress index for alerting the public to stressful weather in Hong Kong. *Meteor. Appl.*, 7: 369–375.

Hajat, S., R.S. Kovats, R.W. Atkinson and A. Haines, 2002: Impact of hot temperature on death in London: a time series approach. *J. Epidemiology and Community Health*, 56(5): 367–372.

Kinney, P.L., M.S. O'Neill, M.L. Bell and J. Schwartz, 2008: Approaches for estimating effects of climate change on heat-related deaths: challenges and opportunities. *Environ. Sci. Pol.*, 11: 87–96.

Anderson, G.B. and M.L. Bell, 2011: Heat waves in the United States: mortality risk during heat waves and effect modification by heat wave characteristics in 43 US communities. *Environ. Health Perspect.*, 119(2): 210–8.

Response: The potential influence of other meteorological factors was considered in our preliminary analyses, but we did not include them in the model because they presented little impact on the association between temperature and mortality. However, to be prudent, we have separately adjusted for relative humidity and wind speed using natural cubic spline with three degree of freedom as a sensitivity analysis (Lines 512-514, Page 15), and have added the results in the revised manuscript (Supplementary Table 11). The effect estimates remained similar after adjusting for relative humidity and wind speed. For instance, under RCP8.5, the heat-related excess mortality were 1.9% (0.2, 3.3) in the 2010s and 5.5% (0.5, 9.9) in the 2090s without adjusting for meteorological factors, 1.9% (0.1, 3.4) in the 2010s and 5.4% (0.7, 9.6) in 2090s with adjustment of relative humidity, and 1.9% (0.2, 3.3) in the 2010s and 5.4% (1.1, 9.3) in 2090s with adjustment of wind speed (Supplementary Table 11).

We agree that the early occurrence of extreme heat events, such as heat waves, might have the greater impact. However, to the best of our knowledge, there is no conclusive evidence that the timing of hot temperature can significantly modify the exposure-response relationship. Our preliminary analysis also supported this

hypothesis (Figure A). The associations between hot temperature and mortality were almost the same before and after taking the timing of hot temperature into account (difference in days between the days with temperature over minimum mortality temperature and the beginning date of hot season-May 1st in each year; for instance, for the hot temperature on 2nd May, the timing of hot temperature for this day is 1, and 3rd May=2, etc.)^{1,2}. Therefore, we did not further consider this issue in our main results, but listed it in the limitations (Lines 344-345, Page 10).

Figure A. The accumulative association between temperature and non-accidental mortality across lag 0-14 days. The red solid line denotes the high temperature; and blue solid line denotes the high temperature with adjusting for the timing of hot temperature; the vertical grey lines denote the minimum mortality temperature; the shaded areas represent 95% CIs.

To sum up, the factors that we included in our model are the most important confounders based on prior knowledge³⁻⁷, and we have also conducted several sensitivity analyses to test the potential influence of other factors. The consistency of our model specifications with previous studies³⁻⁶ and the robustness of our results suggest that our estimates were reasonable and reliable.

References

1. Anderson, G.B. & Bell, M.L. Heat waves in the United States: mortality risk during heat waves and effect modification by heat wave characteristics in 43 US communities. *Environmental health perspectives* **119**, 210-218 (2011).
2. Chen, J., *et al.* The modifying effects of heat and cold wave characteristics on cardiovascular mortality in 31 major Chinese cities. *Environmental Research Letters* **15**, 105009 (2020).
3. Chen, R., *et al.* Association between ambient temperature and mortality risk and burden: time series study in 272 main Chinese cities. *bmj* **363**, k4306 (2018).
4. Gasparrini, A., *et al.* Projections of temperature-related excess mortality under climate change scenarios. *The Lancet Planetary Health* **1**, e360-e367 (2017).
5. Onozuka, D., Gasparrini, A., Sera, F., Hashizume, M. & Honda, Y. Modeling future projections of temperature-related excess morbidity due to infectious gastroenteritis under climate change conditions in Japan. *Environmental health perspectives* **127**, 077006 (2019).
6. Vicedo-Cabrera, A.M., Sera, F. & Gasparrini, A. A hands-on tutorial on a modeling framework for projections of climate change impacts on health. *Epidemiology (Cambridge, Mass.)* (2019).
7. Guo, Y., *et al.* Global variation in the effects of ambient temperature on mortality: a systematic evaluation. *Epidemiology (Cambridge, Mass.)* **25**, 781 (2014).

Reviewers' comments, fourth round:

Reviewer #3 (Remarks to the Author):

I think the authors have addressed all my concerns. I have no further question.

Reviewer #3 (Remarks to the Author):

I think the authors have addressed all my concerns. I have no further question.

Response: We appreciate the reviewer's supportive comments.